# Cancer-Causing Effects of Orthopaedic Metal Implants in Total Hip Arthroplasty

**DOI:** 10.3390/cancers16071339

**Published:** 2024-03-29

**Authors:** Cherry W. Y. Sun, Lawrence C. M. Lau, Jason P. Y. Cheung, Siu-Wai Choi

**Affiliations:** Department of Orthopaedics and Traumatology, School of Clinical Medicine, Faculty of Medicine, The University of Hong Kong, Hong Kong, China; cherrysun21@rcsi.ie (C.W.Y.S.); laucml@hku.hk (L.C.M.L.); cheungjp@hku.hk (J.P.Y.C.)

**Keywords:** orthopaedic implants, carcinogenicity, cobalt, chromium, titanium, total hip arthroplasty

## Abstract

**Simple Summary:**

Total hip arthroplasty is a well-established orthopaedic procedure, the use of which is on the increase worldwide. Since implants used in arthroplasty are usually made of metals including cobalt, chromium and titanium. These metals have since been classified as potentially carcinogenic and may cause cancers in humans. This retrospective study aimed to investigate whether patients with hip implants have a higher risk of developing any type of cancer. The findings will better inform patients and doctors about the risks and benefits of the type of implants and if patients with implants are at a higher risk of malignancy, then patients with implants may benefit from closer medical monitoring.

**Abstract:**

Background: Metal implants have been preferentially used in THA due to its biocompatibility, mechanical stability and durability. Yet concerns have emerged regarding their potential to release metallic ions, leading to long-term adverse effects, including carcinogenicity. This study aimed to investigate the risk of cancer development in patients with orthopaedic metal implants in total hip arthroplasty (THA). Methods: Patients with THA conducted at a local tertiary implant centre from 2001–2008 were linked to the local cancer registry and followed up to the end of 2023. Standardized incidence ratios (SIRs) for cancer incidence and its confidence interval by Poisson distribution were calculated. Survival analysis was depicted using the Kaplan–Meier method, and the log-rank test was used to assess the differences across groups. Results: The study cohort included 388 patients and 53 cancers diagnosed during follow-up, at least 5 years post THA. All-site cancer risks were increased in patients with THA (SIR: 1.97; 95% CI: 1.48–2.46), validated with chi-square analysis (chi-square = 15.2551, N = 100,388, *p* < 0.01). A statistically significant increase in multiple site-specific cancers including haematological cancers were identified. Conclusions: Patients with THA were found to have an increased risk for cancer compared to the general population during a mean follow-up of 16 years.

## 1. Introduction

Total hip arthroplasty (THA) is a well-established orthopaedic procedure that has transformed the lives of millions of individuals worldwide. It is a highly effective and commonly used surgical intervention in alleviating pain and restoring functionality in patients with hip joint pathologies, such as fractures, osteoarthritis, rheumatoid arthritis, and avascular necrosis [1]. In particular, the National Joint Registry Annual Report 2023 stated that 87.9% of primary hip replacements have osteoarthritis as the sole indication for the procedures recorded in the registry [2]. In 2019, the prevalence of osteoarthritis reached 528 million people worldwide, which is an increase of 113% since 1990 [3]. The global prevalence of osteoarthritis is projected to rise due to factors such as ageing populations and increasing rates of obesity and injuries. A consistent annual increase in the number of primary and revision total hip arthroplasty procedures has also been observed over time in developed countries [4]. The Organisation for Economic Co-operation and Development (OECD) reported an average of 172 total hip arthroplasties per 100,000 population in 2021 [5]. Based on epidemiological data in 2019 in the United States, it is forecasted that there will be an increase in THA procedures of 176% by 2040 and 659% by 2060 [6]. Given the increasing utilization of THA worldwide, it is crucial to investigate and thoroughly understand the potential risks associated with these implants.

Over the years, various implant materials made of metals, ceramics, and polymers have been developed, and this has enabled different material combinations for bearing surfaces of joint prostheses, including metal-on-metal (MoM), metal-on-polyethylene (MoP), ceramic-on-ceramic (CoC), and ceramic-on-polyethylene (CoP). Metal implants have been preferentially used and become the standard implant material for THA due to its biocompatibility, mechanical stability, and durability, where MoP surface bearings are the most commonly used combination [7]. Orthopaedic metal implants used in THA are typically composed of alloys containing cobalt, chromium, and titanium, with cobalt-chromium alloys being one of the major hip implant materials composing the femoral stem and articulating head [8]. While these metals are considered biologically inert and have superior wear and corrosion resistance, concerns have emerged regarding their potential to release metallic ions that can be incorporated into biological molecules and particulate debris (due to degradation and wearing of materials) into the surrounding tissues, leading to long-term adverse effects, including carcinogenicity.

In terms of carcinogenicity classification, cobalt was up-classified to category 1B (presumed to have carcinogenic potential for humans by the European Chemicals Agency (ECHA) in 2019 primarily based on the cancer risk identified in rodent metallic cobalt inhalation studies. Although the carcinogenic risks of cobalt chromium alloys in medical devices were not evaluated, a separate risk assessment of cobalt exposure from medical devices is required under the European Medical Device Regulations (MDR) [9]. In 2023, the International Agency for Research on Cancer (IARC) reported that cobalt metal and soluble cobalt (II) salts were being classified into group 2A (probably carcinogenic to humans) and cobalt (II) oxide were being classified into group 2B (possibly carcinogenic to humans) [10]. Cobalt-chromium alloys are widely used in orthopaedic hip implants; therefore, the up-classification warranted further investigations regarding the carcinogenicity of metals in THA.

Although the underlying mechanisms by which orthopaedic metal implants may contribute to carcinogenesis are not yet fully understood, several hypotheses have been proposed. One of the primary concerns is the genotoxic potential of metal ions, which have been shown to induce DNA damage and genomic instability, potentially leading to the development of malignancies [11]. In addition, chronic inflammation induced by the presence of metal debris may play a crucial role in the initiation and promotion of carcinogenesis. Metal ion levels were found to have increased after THA in blood and urine samples of patients [12,13,14]. Studies have shown that released particles can induce a localized inflammatory response, leading to adverse tissue reactions such as metallosis, osteolysis, and pseudotumors [15,16,17]. Additionally, they have the potential to migrate to distant sites and cause systemic adverse effects involving multiple organs, raising questions about their long-term safety [18]. Numerous retrospective cohort studies have been conducted to investigate the association between orthopaedic metal implants and cancer development. Some studies have reported an increased risk of certain types of cancer, including lymphoma, prostate cancer, and melanoma [19,20,21]. In particular, higher prostate and skin cancer risk was identified in a MoM cohort compared to the non-MoM cohort [19]. However, conflicting findings have been observed in other studies, suggesting no increase in all-site and site-specific cancer risks associated with THA. These discrepancies may be attributed to variations in study design, patient populations, implant characteristics, follow-up duration, and other confounding factors. Besides the material of the bearing couples, cementation and primary reason for THA might also influence the long-term surgical outcomes post THA. A slight elevated cancer risk has been identified in the patient group with uncemented THA compared to that without cementation [22].

Understanding the potential carcinogenicity of orthopaedic metal implants in total hip arthroplasty is of paramount importance for both patients and healthcare providers considering the high worldwide prevalence of implant use. It is imperative to comprehensively assess the potential risks associated with hip prostheses, especially in light of their long-term exposure within the human body. Yet, currently available epidemiological studies are inconclusive as to whether metallic THA is associated with an increase in cancers. In the present study, we investigated the risk of all-cause and site-specific cancer in patients who underwent total hip arthroplasty in 2001–2008 by mining data from the Clinical Management System of a tertiary hospital in Hong Kong and the Hong Kong Cancer Registry. The primary aim of the study was to determine if there were any differences in the long-term cancer risk in patients with THA according to bearing surfaces and cementation of THA when compared to the general population in order to better inform clinicians and patients about the need for monitoring post THA surgery.

## 2. Methodology

### 2.1. Study Population

The study population was drawn from the database of the Clinical Management System at Queen Mary Hospital (Institutional Review Board approval number UW 13–176). Patients undergoing primary or revision total hip arthroplasty during 2001–2008 were included, with this period being chosen to enable an adequate timeframe for cancer development. The inclusion criteria also include patients with available medical records and data on the type of metal implants used and a minimum follow-up period of 5 years to allow for the assessment of carcinogenicity. Each member of the study cohort was followed up from the date of their primary or first revision total hip arthroplasty until cancer, emigration, death, or 31 December 2023 (study census), whichever came first (Figure 1).

There is no currently available evidence suggesting the required time period from metal implant procedures to cancer development, but cancers are known to have long latent periods of at least 5 years for development or detection. Hence, only primary cancers diagnosed during the follow-up period 5 years post total hip arthroplasty were counted as observed cases. Cancers occurring before and 0–5 years after each patient’s procedure were not regarded as relating to the implant. For comparison, the all-site and site-specific cancer incidences of the general population were obtained from the Hong Kong Cancer Registry.

Sub-cohorts were stratified based on the (1) procedure: primary or revision THA; (2) bearing surfaces: MoM, MoP, CoC, or CoP bearings; (3) types of cementation: cemented or non-cemented; (4) reasons for THA: polyethylene wear, osteolysis and loosening, dislocation and recurrent subluxation, avascular necrosis and non-union, fracture, or sepsis and infection.

Diagnoses of cancer were classified according to the International Classification of Diseases version 10 (ICD-10) [23]. The analysis was stratified by cancer types: Head and Neck (C00–14); Oesophagus, Stomach, Small intestine, and Colon (C15–18); Rectum (C20); Liver (C22); Gallbladder (C23); Pancreas (C25); Lung (C34); Heart (C38); Bone and articular cartilage (C40–41); Skin other than melanoma (C44); Breast (C50); Cervix and Corpus uteri (C53–54); Gestational Trophoblastic Neoplasm (C58); Prostate (C61); Urological (C64–68); Thyroid (C73); and Haematolymphatic cancers (C81–C96) according to ICD-10.

### 2.2. Statistical Methods

Standardised incidence ratios (SIRs) and their confidence intervals by Poisson distribution were calculated based on the ratio of age-specific rates of cancer observed in our study cohort to observed age-specific rates of cancer in the general population of Hong Kong with the data from the Hong Kong Cancer Registry. Survival analysis was conducted to compare the cancer incidence rates between patients with THA and the general population. The duration from the date of total hip replacement to the occurrence of cancer or the end of the study period was considered the survival time for patients with THA. The general population was followed up for the same duration. Survival curves were generated using the Kaplan–Meier method, and the log-rank test was used to assess the differences across groups. IBM-SPSS version 29 was used for the statistical analyses. In this study, *p*-values < 0.05 were considered statistically significant.

## 3. Results

### 3.1. Characteristics of the Study Population

The study cohort followed 388 patients who had received THA between 2001–2008, for the SIR calculations (193 patients who underwent primary THA and 195 patients who underwent revision THA). The median follow-up period was 16 years (25th percentile (Q1) = 9; 75th percentile (Q2) = 19; range 0–22). No patients were lost to follow-up. The male to female ratio was 1:1.31, (females represented 56.7% of cohort). The median patient age at surgery was 61 years (Q1 = 50; Q2 = 73; range 19–94) (66.0 years for females and 54.5 years for males). Notably, 168 patients had died before the project census (Table 1). The sub-cohorts for bearing surface analysis included two patients with MoM bearings, 366 with MoP, 10 with CoC, and 4 with CoP bearings. The sub-cohorts for the use of bone cement included 264 patients with non-cemented THA and 50 patients with cemented THA components. The sub-cohorts for indications for THA revision included 50 patients due to polyethylene wear, 106 due to osteolysis and loosening, three due to dislocation and recurrent subluxation, 12 due to avascular necrosis and non-union, 7 due to fracture, and 14 due to sepsis or infection (Table 2).

In the follow-up period, 78 cancers were observed in this cohort, 53 of which were diagnosed five years after THA (Table 3). The mean time from THA to cancer diagnosis was seven years (range 0–21), when patients with cancer diagnosed within five years of THA were removed from the analysis, the mean time from THA to cancer diagnosis was 13 years (range 6–21) (Figure 2). Cancer incidence was the highest among the 80 year old age group, and all cases were observed in the sub-cohort with MoP bearing surfaces. None of the cancer cases were diagnosed in patients under 50 years old. Over half of the cancer incidences were identified in patients with revision THA (revision required due to polyethylene wear and osteolysis and loosening) (Table 3).

### 3.2. Statistical Analysis

Regarding all-site cancer risk in the whole cohort, cancer risk was higher in patients with THA compared to the general population (SIR 1.97, 95% CI: 1.48–2.46). Chi-square of cancer incidence in patients with THA versus the Hong Kong general population was Chi-square = 15.2551 (N = 100,388), *p* < 0.01, showing a statistically significant association of cancer incidence with THA. Regarding site-specific cancer risk in the study population, six types of cancers including breast (Chi-square = 7.1814, *p* < 0.01), haematological (Chi-square = 6.081, *p* < 0.05), head and neck (Chi-square = 27.4421, *p* < 0.001), oesophageal (Chi-square = 9.49, *p* < 0.01), and small intestine (Chi-square = 4.6082, *p* < 0.05) cancers were identified as having statistically significant increased incidence rates. The site-specific cancer risk of prostate cancer (Chi-square = 4.6082, *p* < 0.05) in patients with THA was statistically significantly lower than in the general population (Figure 3).

## 4. Discussion

Total hip arthroplasty has been named “the operation of the century” due to its high success rate, long survivorship, and satisfactory clinical outcome. Among the materials used in orthopaedic implants, metals have gained widespread popularity due to their high strength, durability, and biocompatibility. Chromium, cobalt, and titanium have revolutionized the field of orthopaedics and are extensively utilized metals in orthopaedic implants due to their favourable mechanical properties, resistance to corrosion, superior wear resistance, and biocompatibility. However, concerns have been raised regarding the potential carcinogenicity of metals used in orthopaedic implant. While orthopaedic implants have greatly improved patient outcomes, long-term exposure to metals may pose potential health risks. Assessment of the potential carcinogenicity of metals used in implants is a critical aspect that needs to be addressed to ensure patient safety.

Our data show a statistically significant increased all-site cancer incidence of 2.93% in patients with THA compared to that of the general population in matched age groups. Additionally, a statistically significant increase in cancer incidence was identified in the study population for breast, haematological, head and neck, oesophagus, and small intestinal cancer. It is worth noting that prostate cancer incidence was statistically significantly lower in the study population than that of the general population.

The data presented here showing an increased standardised incidence ratio and increases in all-site and site-specific cancer incidences in patients with THA are consistent with previous findings. A study conducted with data from the Swedish Knee Arthroplasty Register observed an increased overall risk of cancer and risk of several cancer types compared with the general population [20]. An excess cancer incidence of 10–26% compared to the general population was identified in the study. A Scottish study also found a significant increase in cancer incidence in THA patients [24]. These findings are further supported by an Australian nationwide linked registry cohort analysis of 167,837 patients, demonstrating an increased all-site cancer incidence in patients with THA [21]. However, multiple studies in England, Finland, and Sweden show data to the contrary [19,25,26,27,28]. In terms of site-specific cancer risks, various studies noted an increased risk of haematopoietic cancers among patients with THA, yielding comparable results [20,24,29,30,31]. An increased risk of prostate cancers was recognised in numerous studies, but contradictory results have been obtained in this study.

As demonstrated in the Kaplan–Meier curve, patients who received THA at a younger age tended to have a longer procedure-to-cancer diagnosis period. There was no consistent time period from THA procedure to cancer diagnosis observed across age groups. Considering this with the demographic properties of the study population and that the oldest age group yielded the majority of the cancer cases, it is possible to speculate that the observed difference in incidence between patients with THA and the general population could be due to factors including demographic differences or other confounding factors.

A strength of this current study is the long-term follow-up, with a median of 16 years. Latency of cancer holds significance in determining its cause. The Centre for Disease Control and Prevention determined the minimum latencies for all solid cancers as four years and haematopoietic cancers as 0.4 years [32]. Although the time period at which a cancer can be directly attributable to prostheses remains uncertain, there are indications that any increased cancer risk observed early in the follow-up period could be influenced by factors such as detection bias or disease-related causes. To minimise bias, cancers diagnosed in the first five years of THA were excluded from this analysis. One of the weaknesses of this study is the small population size included in this study, with higher risk of selection bias and confounding factors. However, the data from which this study is based on is one of the largest joint replacement centres in the territory, as well as being the largest centre for cancer patients. It is possible for patients to be given a joint replacement at this centre but be followed-up somewhere else due to convenience, but as long as the patient is a resident in the territory, their medical records would be entered into a territory-wide centralised system. It is unlikely that this study lost many patients to follow-up unless the patient left the country. In addition, the cancer rates in joint replacement populations may be increased due to surveillance bias due to opportune discovery while being followed up on the joint replacement.

## 5. Conclusions

This study identified a higher cancer burden in patients with THA compared to the general population. Adverse effect surveillance of materials and prostheses and the re-evaluation of implant materials are warranted to ensure patient safety. The results from this study have shown that there is a need for further investigations into the mechanisms underlying the observed associations, the specific types of implants involved, or the impact of other factors, such as patient characteristics or implant design.

## Figures and Tables

**Figure 1 cancers-16-01339-f001:**
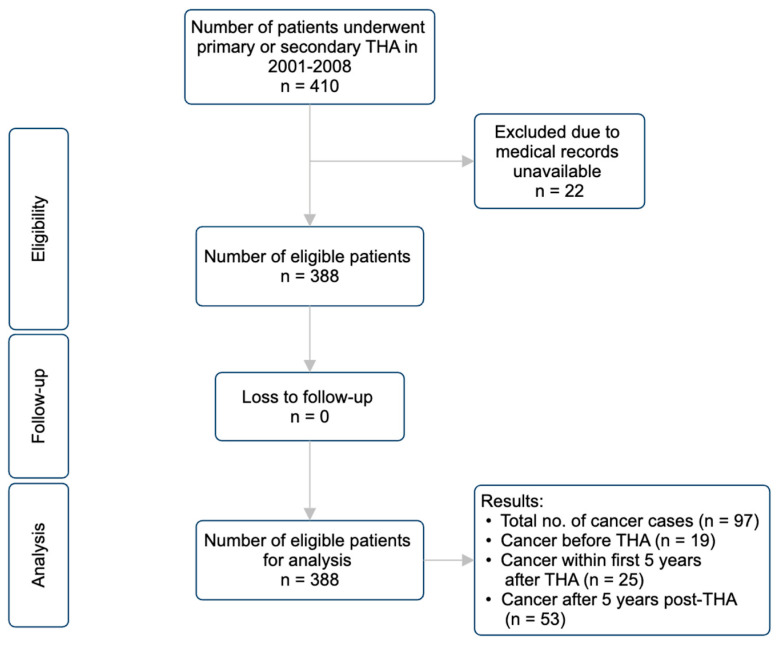
Study procedure.

**Figure 2 cancers-16-01339-f002:**
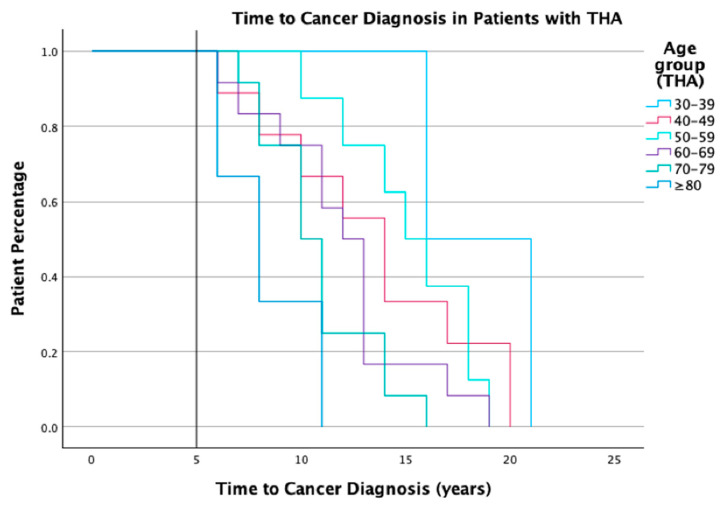
Kaplan–Meier analysis of time to cancer diagnosis in patients with THA.

**Figure 3 cancers-16-01339-f003:**
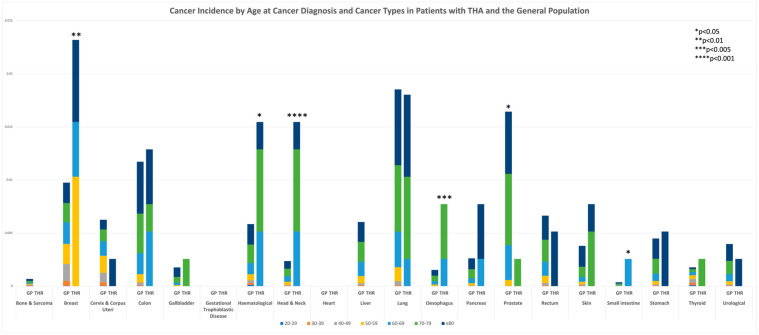
Cancer incidence by age at diagnosis and cancer types versus the general population in Hong Kong. All-site cancer incidence in general population: 10.47%; THA: 13.40%.

**Table 1 cancers-16-01339-t001:** Number of patients with total hip arthroplasty stratified by sex, age at operation, up to the end of 2023.

Age (Years)	Men				Women				All			
	Total (n)	Alive (n)	Dead (n)	Age at Death (Years)	Total (n)	Alive (n)	Dead (n)	Age of Death (Years)	Total (n)	Alive (n)	Dead (n)	Age of Death (Years)
0–29	10	8	2	0	3	3	0	0	13	11	2	0
30–39	12	9	3	1	11	10	1	0	23	19	4	1
40–49	40	27	13	5	19	16	3	0	59	43	16	5
50–59	42	29	13	6	48	34	14	4	90	63	27	10
60–69	30	15	15	18	44	28	16	5	74	43	31	23
70–79	25	12	13	14	64	21	43	18	89	33	56	32
≥80	9	3	6	21	31	5	26	76	40	8	32	97
Total	168	103	65		220	117	103		388	220	168	

**Table 2 cancers-16-01339-t002:** Number of patients in sub-cohorts (bearing surfaces, cementation, reasons for THA) by age at operation.

Group	Age Group							
Subgroups	0–29	30–39	40–49	50–59	60–69	70–79	≥80	Total
All	13	23	59	90	74	89	40	388
THA								
Primary	11	12	34	47	39	40	10	193
Revision	2	11	25	43	35	49	30	195
Bearing surface								
MoM	0	0	0	1	0	1	0	2
MoP	7	22	54	84	72	87	40	366
CoC	6	0	2	2	0	0	0	10
CoP	0	1	2	0	1	0	0	4
Cementation								
Uncemented	11	15	42	65	54	62	15	264
Cemented	1	1	5	4	5	12	22	50
Reasons for THA								
Wear	1	6	8	14	11	9	1	50
Osteolysis	0	3	14	27	16	27	19	106
Dislocation	0	2	0	0	0	0	1	3
AVN	0	0	0	1	2	6	3	12
Fracture	0	0	0	0	2	3	2	7
Sepsis	1	0	2	1	3	4	3	14

**Table 3 cancers-16-01339-t003:** Number of patients with total hip arthroplasty stratified by age at operation and cancer incidence up to the end of 2023 and cancer incidence in sub-cohorts (bearing surfaces, cementation, reasons for THA) by age at cancer diagnosis.

Cancer Incidence	Age Groups							
Subgroups	0–29	30–39	40–49	50–59	60–69	70–79	≥80	Total
No Cancer	11	20	46	74	53	63	31	298
Cancer developed before THA	0	0	5	4	5	2	3	19
Cancer developed ≤5 years after THA	1	2	4	5	3	7	2	24
Cancer developed >5 years after THA	1	1	4	7	13	17	4	47
Total	13	23	59	90	74	89	40	388
Subgroups of Cancer Developed >5 Years after THA								
THA								
Primary	0	0	0	6	2	8	8	24
Revision	0	0	0	0	8	9	12	29
Bearing surface								
MoM	0	0	0	0	0	0	0	0
MoP	0	0	0	6	10	16	21	53
CoC	0	0	0	0	0	0	0	0
CoP	0	0	0	0	0	0	0	0
Cementation								
Uncemented	0	0	0	6	5	11	11	33
Cemented	0	0	0	0	0	2	4	6
Reasons for THA								
Wear	0	0	0	0	4	1	3	8
Osteolysis	0	0	0	0	4	4	4	12
Dislocation	0	0	0	0	0	0	0	0
AVN	0	0	0	0	0	0	3	3
Fracture	0	0	0	0	0	0	1	1
Sepsis	0	0	0	0	0	2	0	2
Total	0	0	0	5	11	16	21	53

## Data Availability

The data presented in this study are available on request from the corresponding author.

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
