# Peer review of "Cancer-Causing Effects of Orthopaedic Metal Implants in Total Hip Arthroplasty"

_cancers, 2024, doi:10.3390/cancers16071339_

Round 1
Reviewer 1 Report
Comments and Suggestions for Authors
The article submitted for review is interesting. Statistical data based on a large group of patients, which is a big advantage. It has been found that the risk of cancer is increased in patients with THA. According to the reviewer, the research is well documented, described and extensive (over the last 16 years). The article is worth publishing in its current form, after minor linguistic and editorial corrections, such as the legend in Fig. 1, which is difficult to read.
Comments on the Quality of English LanguageThe article submitted for review is interesting. Statistical data based on a large group of patients, which is a big advantage. It has been found that the risk of cancer is increased in patients with THA. According to the reviewer, the research is well documented, described and extensive (over the last 16 years). The article is worth publishing in its current form, after minor linguistic and editorial corrections, such as the legend in Fig. 1, which is difficult to read.
Author Response
Reviewer 1
The article submitted for review is interesting. Statistical data based on a large group of patients, which is a big advantage. It has been found that the risk of cancer is increased in patients with THA. According to the reviewer, the research is well documented, described and extensive (over the last 16 years). The article is worth publishing in its current form, after minor linguistic and editorial corrections, such as the legend in Fig. 1, which is difficult to read.
Thank you for your comments, the legend to figure 1 (which has now become figure 2) has been changed to “Cancer incidence by age at diagnosis and cancer types versus the general population in Hong Kong”
Reviewer 2 Report
Comments and Suggestions for Authors
The work is very interesting in an complicated issue.
I need a clarification on a phrase in Results:
-The study cohort yielded 5246 person years. ¿What mean it?.
Another questions is the number of cases. It is a tertiary hospital, and the number of cases for that hospital in that period of time is low. This catches my attention.
And the results are surprising. Of 388 people, 78 with cancer, it is 20%. It is high. What the opinion of the authors for that.
Author Response
Reviewer 2
The work is very interesting in an complicated issue.
I need a clarification on a phrase in Results:
-The study cohort yielded 5246 person years. What mean it?. Thank you for your comments, the person years has been removed from the manuscript.
Another questions is the number of cases. It is a tertiary hospital, and the number of cases for that hospital in that period of time is low. This catches my attention. Thank you for raising this matter, the reason for this is that this analysis only covers patients with total hip implants, rather than other implants including knee. We decided to study a homogenous dataset of only hip implant patients instead of all orthopaedic implants, which explains the low number of patients.
And the results are surprising. Of 388 people, 78 with cancer, it is 20%. It is high. What the opinion of the authors for that. Thank you for raising this issue. Since the total hip replacements are (usually) conducted in the older population, the percentage later developing cancer is in line with, and slightly higher than the general all-age population.
Reviewer 3 Report
Comments and Suggestions for Authors
Dear Authors,
I give you the following question to address in your manuscript: It enhances the researcher's understanding and readability.
1. What are the primary reasons for the widespread use of metal implants in total hip arthroplasty (THA)?
2. What specific concerns have arisen regarding the potential release of metallic ions from orthopedic metal implants, and how might these concerns affect long-term patient health?
3. What is the primary objective of this study regarding the cancer risk associated with orthopaedic metal implants used in THA?
4. How were patients included in the study cohort, and what was the duration of follow-up for these patients?
5. What methodology was employed to assess cancer risk among patients with THA, and what statistical measures were used to analyze the data?
6. What were the key findings regarding cancer incidence among patients with THA during the follow-up period?
7. Were there any specific types of cancer that showed a statistically significant increase in incidence among patients with THA compared to the general population? If so, which ones?
8. What conclusions can be drawn from the study regarding the association between THA and cancer risk, and what implications might these conclusions have for clinical practice and patient care?
9. In the introduction section, please add a schematic diagram to address your research.
Best Regards
Author Response

(The authors gave the same response as above.)
